# Ultrasound Sensor-Based Wireless Power Transfer for Low-Power Medical Devices

**Mustafa F. Mahmood, Saleem Latteef Mohammed and Sadik Kamel Gharghan ***

Department of Medical Instrumentation Techniques Engineering, Electrical Engineering Technical College, Middle Technical University, Baghdad 10001, Iraq
* Correspondence: sadik.gharghan@mtu.edu.iq; Tel.: +964-7736393936

**Abstract:** Ultrasonic power transfer (UPT) is a promising method for wireless power transfer technology for low-power medical applications. Most portable or wearable medical devices are battery-powered. Batteries cannot be used for a long time and require periodic charging or replacement. UPT is a candidate technology for solving this problem. In this work, a 40-KHz ultrasound transducer was used to design a new prototype for supplying power to a wearable heart rate sensor for medical application. The implemented system consists of a power unit and heart rate measurement unit. The power unit includes an ultrasonic transmitter and receiver, rectifier, boost converter and super-capacitors. The heart rate measurement unit comprises measurement and monitoring circuits. UPT-based transfer power and efficiency were achieved using 1-, 4- and 8-Farad (F) super-capacitors. At 4 F, the system achieved 69.4% transfer efficiency and 0.318 mW power at 4 cm. In addition, 97% heart rate measurement accuracy was achieved relative to the benchmark device. The heart rate measurements were validated with statistical analysis. Our results show that this work outperforms previous works in terms of transfer power and efficiency with a 4-cm gap between the ultrasound transmitter and receiver.

**Keywords:** Arduino; heart rate sensor; nRF24L01; super-capacitors; transfer efficiency; ultrasonic power transfer

## 1. Introduction

Implantable biomedical devices such as blood pressure monitors, cardiac defibrillators, electrocardiograms, electromyography, thermometers, pacemakers, neural stimulators, heart rate sensors, and glucose meters are commonly used to improve the quality of life of millions of patients [1]. Ultrasound can be defined as cyclic sound pressure with a frequency greater than the upper limit of human hearing. Some animals, for example, bats and dolphins, use ultrasonic waves for locating prey and obstacles. In recent years, the range of ultrasound applications open to humans has expanded considerably, and includes the medical, chemical, industrial and military fields [2]. An ultrasonic sensor is an audio sensor, and can be divided into three main categories: Transmitter, receiver and transceiver. To improve medical implant devices, several investigators [3,4] have used ultrasonic waves to optimize transfer efficiency and increase transfer distances in percutaneous coronary intervention. In [5], this was done to alter the operating frequency to optimize transfer distances between receiver and transmitter sensors. Other researchers have used a thermal and vibration hybrid based on ultrasonic sensors to develop transfer power [6]. One study used microelectronics devices to optimize transfer efficiency and compared it with inductive power transfer. Ultrasonic wave–based transfer power is an alternative method to active wearable medical sensors as compared with the electromagnetic method [7]. Additional components were used in medical implant devices, such as matching layers, to improve transfer efficiencies [8].

The present study involved an ultrasonic power transfer (UPT) system consisting of a transmitter (Tx) ultrasonic sensor, receiver (Rx) ultrasonic sensor, bridge rectifier, boost converter and load (super-capacitor). Furthermore, the system contains a measurement unit and monitoring unit. The measurement unit involves a microcontroller heart rate sensor and wireless protocol nRF24L01. The monitoring unit has a microcontroller, wireless protocol nRF24L01 and a laptop. The main objective of the present study is to optimize the transfer distance, power and efficiency.

UPT-based transfer power and efficiency were achieved using 1-, 4- and 8-Farad (F) super-capacitors. The transfer power was adequate for supplying a wearable heart rate sensor. The system was tested using transfer distances of 1–10 cm. Several super-capacitors were considered to select the best capacitor that could yield high transfer power and efficiency relative to the other capacitors. At 4 F, the system achieved 69.4% transfer efficiency and 0.318 mW power at 4 cm. In addition, 97% heart rate measurement accuracy was achieved relative to the benchmark device. The heart rate measurements were validated with statistical analysis. Our results show that, with a 4-cm gap between the ultrasound transmitter and receiver, this work outperforms previous works in terms of transfer power and efficiency.

The contribution of this paper can be summarized as follows:

- Designed and implemented lightweight UPT.
- The ultrasound transducer was designed and implemented to supply adequate power to a wearable heart rate sensor. We achieved transfer power and efficiency at different distances with several super-capacitors.
- Improved transfer power and efficiency between transmitter and receiver relative to related previous works.
- Heart rate monitoring (HRM), which includes heart rate sensor, microcontroller and wireless protocol (i.e., nRF24L01), was successfully provided by UPT-based power.
- Heart rate measurement was validated with a benchmark (BM) based on statistical analysis.

The rest of the paper is as follows. Related work such as that for the performance metrics of the transfer energy is discussed in Section 2. Section 3 presents the proposed system design. Section 4 describes and introduces the system experiment configurations. Validation of the heart rate monitoring system based on statistical analyses is described in Section 5. Section 6 highlights the overall results and discussion of the UPT system. The results of the proposed system are compared with that of previous related works in Section 7. Finally, conclusions with trends for future work are drawn in Section 8.

## 2. Related Works

This section highlights previous works using ultrasonic sensors to generate power for vital signs patient monitoring systems. Islam et al. [3] proposed a vibration to optimize transfer distance and efficiency for implantable devices in percutaneous coronary intervention. The system operates at a frequency of 14 MHz. The system consists of ultrasonic power, polyvinylidene fluoride (PVDF), rectifier and autonomous active stent (AAS) sensor. The experimental results showed that the power transfer efficiency based on AAS and PVDF was approximately 14.8%, while power transfer efficiency of 11.5% was achieved with lead zirconate titanate. The output voltage was 200 mV at the short distance of 2.5 mm and low resonant frequency. At the same distance, the transfer power efficiency of the system using AAS and PVDF was better than the lead zirconate titanate system. The system has several advantages, such as small size, wearability and low cost. However, it has a short transfer distance. Shi et al. [4] designed and implemented a vibration for medical implant devices to improve transfer power and efficiency based on the piezoelectric material sensor harvesting technique. The system operated at frequencies of 250–240 kHz. The system includes the ultrasonic transmitter/receiver and load. The experimental results revealed that the output voltage of the device was 0 dB at 0 cm, whereas this decreased as the distance increased. The transfer power was improved from 12.0 nW to

77.1 nW at 1 cm based on the piezoelectric ultrasonic energy harvester. The piezoelectric ultrasonic energy harvester system was better than that without it in terms of transfer power efficiency at the same distance. The system had several advantages such as small size, wearability and low cost. However, it has a short transfer distance.

Vihvelin et al. [5] presented a vibration for medical implant devices to improve transfer efficiency. The system was powered based on the energy harvesting technique (i.e., ultrasonic wave). The system operates at 1.2–1.4 MHz and includes a piezoelectric transmitter/receiver and load. The authors tested two types of media. The experimental results showed that the transfer efficiencies and distance between transmitter and receiver differed based on frequency and media. In water, the transfer efficiency was 45% at transfer distances of 5.9 mm and 6.1 mm at 1.35 and 1.29 MHz, respectively. In air, transfer efficiency was 48% at 4 mm at 1.02–1.60 MHz. The results show that the corresponding transfer efficiency for varying frequencies was 35–100% whereas it was 8–25% for fixed frequencies. The advantages of the system were its low cost, wearable device and small size. Nevertheless, it has a short transfer distance. Semsudin et al. [6] merged two harvesting techniques (vibration and thermal) for supplying power to micro biomedical devices. The operating frequency of the system is 2 kHz. The system includes triboelectric generators/thermal, rectifier, DC–DC converter and load. The authors tested three inductance variation values (i.e., 0.4, 0.9, 1.2 µH). Experimental results revealed that the output voltages were 2 V, 3.2 V and 4 V, for 0.4 µH, 0.9 µH and 1.2 µH, respectively, using a 0.3 V vibration source. When a 0.2 V thermal source was used, the output voltages were 2.2 V, 3 V and 4 V for 0.9 µH, 1.7 µH and 2.9 µH, respectively. When thermal and vibration were hybridized, the output voltages were 2 V, 3.5 V and 4 V for 0.3 µH, 0.6 µH and 0.9 µH, respectively, based on 0.5 V input voltage. The proposed system was low cost and wearable. Nevertheless, it had high power consumption.

Meng et al. [7] designed a wearable sensor for implantable microelectronic devices to optimize transfer efficiency. The wearable sensor was powered based on the energy harvesting technique (i.e., vibration of the ultrasonic sensors). The system operated at 1 MHz, 10 MHz and 15 MHz, and included an ultrasonic transmitter/receiver, rectifier, regulator and load. The authors tested three types of frequencies, and the experimental results showed that transfer efficiency was 6%, 0.03% and 0.15% at 3 cm and 5%, ~0.043% and ~0.06% at 8 cm for 1 MHz, 10 MHz and 15 MHz, respectively. In addition, they showed that inductive transfer efficiency was better than ultrasonic at 3 m, being 0.27% and 0.03%, respectively; however, at 8 cm, ultrasonic had better transfer efficiency than inductive, being 0.04% and 0.0025%, respectively. The system has several advantages, such as small size, long distance, wearability and low cost. However, it has poor transfer efficiencies. Miao et al. [8] proposed a vibration technique to supply power to medical implant devices. The transfer distance and efficiency were improved using ultrasonic waves. The system operates at 1.1–1.8 MHz and includes a piezoelectric transmitter/receiver, rectifier, regulator and load. The authors examined two types of piezoelectric and the experimental results revealed that the transfer efficiency between the piezoelectric transmitter and receiver was 2.11% and 1.15% for 1.8 MHz and 1.4 MHz, respectively, at 3 cm. However, the transfer efficiencies were improved when the matching layer was considered at 1.1 MHz; then, the transfer efficiencies were 65% and 18% at a transfer distance of 3 and 6 cm, respectively. The advantages of the system are the wearable device, small size, safety and low cost. Nevertheless, it has a short transfer distance.

The previous works are summarized in Table 1. The table compares the performance metrics of the previous studies in terms of operating frequency, harvesting technique, transfer power and efficiency at different distances.

**Table 1.** Performance metrics of previous works.

| Ref. | Objective | Operating Frequency (MHz) | Harvesting Technique | Implementation Environment | Application | Transfer Distance (cm) | Transfer Efficiency (%) | Power (W) |
|------|-----------|---------------------------|----------------------|----------------------------|-------------|------------------------|-------------------------|-----------|
| [3] | Optimize transfer distance and efficiency | 14 | Vibration | Experimental | Percutaneous coronary intervention | 3 | 14.8 | N/A |
| [4] | Optimize transfer power | 0.24–0.25 | Vibration | Experimental | Implant devices | 1 | N/A | 7.71 ×10⁻⁸ |
| [5] | Improve transfer efficiency | 1.2–1.4 | Vibration | Experimental | Implant devices | 0.61 | 45 | N/A |
| [6] | Improve output voltage | 0.2 | Vibration and thermal | Simulation | Micro biomedical devices | N/A | N/A | N/A |
| [7] | Improve transfer efficiency | 1, 10, 15 | Vibration | Experimental | Implant devices | 4 | 5.5 @ 1 0.035 @ 10 0.087 @ 15 | N/A |
| [8] | Improve transfer efficiency | 1.1–1.8 | Vibration | Experimental | Implant devices | 4 | 1.14 | N/A |

N/A: not available.

## 3. System Model

The proposed UPT shown in Figure 1 consists of two parts: A transmitter and receiver. The transmitter includes a function generator and the Tx. The receiver comprises the Rx, bridge rectifier, boost converter and load (super-capacitor). The HRM consists of the measurement unit and monitoring unit. The measurement unit involves an Arduino (Nano) microcontroller, heart rate sensor and wireless protocol nRF24L01. The monitoring unit has an Arduino Mega 2560 microcontroller, wireless protocol nRF24L01 and a laptop.

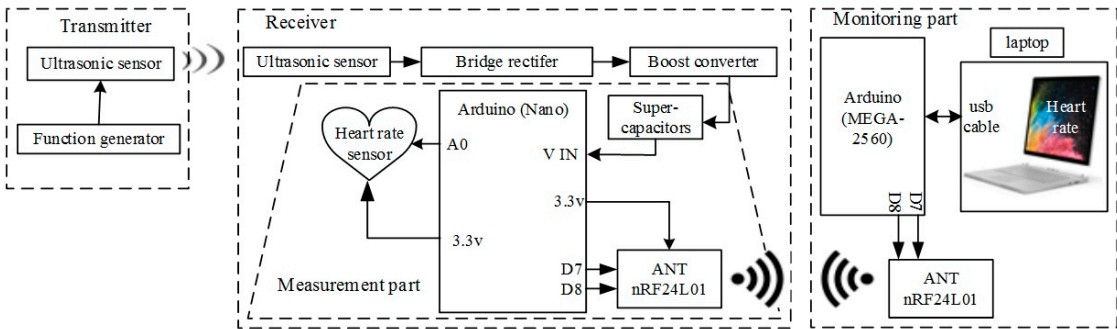

**Figure 1.** Block diagram of use of ultrasonic power transfer (UPT)-based heart rate sensor.

### 3.1. Transmitter Sensor

The transmitter uses a function generator and Tx. The function generator for generating an electrical signal to the ultrasonic transmitter. Tx converts electrical signals to ultrasound signals.

### 3.2. Receiver Sensor

The receiver comprises Rx, rectifier, boost converter and load (super-capacitor). The receiver sensor converts ultrasound signals to electrical signals. The signal from this stage is alternate. The bridge rectifier is used to convert AC voltage to DC voltage. Therefore, we used the LTC-3588 model [9]. The boost converter is used because the voltage from the bridge is low. The load acts as a power supply for the device. The super-capacitor is a high-farad (F) component; it consists of two plates to store power. At the same time, we used a power bank or power source. In addition, we studied the time for

capacitor charging and discharging in Equations (1)–(3) for charging the capacitor. These equations explain the mechanisms for both charging times [10].

$$V_c = V_s\left(1 - e^{-\frac{t}{\tau}}\right) \tag{1}$$

$$I_c = I_s\left(1 - e^{-\frac{t}{\tau}}\right) \tag{2}$$

$$\tau = RC \tag{3}$$

where $V_c$ and $I_c$ are voltage and current, respectively, at a specific time. $V_s$ and $I_s$ are the voltage and current, respectively, at the source; $\tau$ is the time constant, $R$ is the resistor and $C$ is the capacitor.

The Arduino Nano microcontroller in the receiver is connected to the heart rate sensor. The heart rate is the number of heartbeats in a specific period, while pulse changeability refers to the variety in the intervals between continuous heartbeats. Besides, the heart rate sensor was linked to low voltage for decreased power consumption, was small, i.e., 16-mm diameter, and low cost [11,12]. The wireless protocol nRF24L01 is an independent module; it has low-power consumption, low cost, is small and has a long data-sending distance, i.e., up to 100 meters; it uses a 2.4-GHz radio frequency. It has many applications, such as wireless computer peripherals, wireless data communication, industrial sensors, and ultra-low-power sensor networks. In the present model, we enabled control of media access, address, information and error detections and retries to secure data and receive it [13,14].

### 3.3. Monitoring of Heart Rate Information

The monitoring unit involved an Arduino Mega 2560 microcontroller, wireless protocol nRF24L01 and a laptop. The microcontroller is based on the ATmega 2560 processor [15]. It is connected to the wireless protocol on the computer to show the result. Further, the Arduino Nano microcontroller was included in the transmission unit because it is small, targeted at low-power consumption, lightweight and low cost [16]. In addition, it is programmed using C++ language [17]. Wireless protocol nRF24L01 and the laptop have been explained in the measurement part. The laptop is used to show the results.

## 4. Experiment Configuration

The ultrasonic sensor was a HC-SR04 model. The Tx ($45 \times 20 \times 15 \text{ mm}^3$) converted electrical signals to ultrasound signals; the input voltage was square wave, 10 $V_{\text{peak-to-peak}}$ at 40 KHz from the function generator (model VC2002) (Figure 2a). The Rx ($45 \times 20 \times 15 \text{ mm}^3$) converted the ultrasound signal to an electrical signal with amplitude and frequency (Figure 2b) [18].

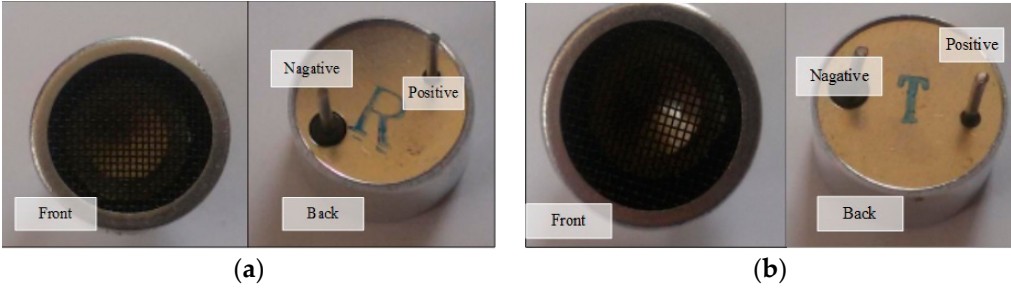

**Figure 2.** The front and back faces of an ultrasonic sensor; (**a**) transmitter sensor and (**b**) receiver sensor.

The UPT depended on the distance between the Tx and Rx. The next stage was converting AC to DC voltage using the rectifier (model LTC-3588, $2 \times 1.25 \text{ cm}^2$); this model functioned at low voltage. The Rx received the voltage and frequency to attach to the bridge to rectify input elements at piezoelectric 1 and piezoelectric 2. The output voltage was direct from the element at VIN pin. Subsequently, the voltage for that was low, so we used a boost converter to increase the voltage. In the

present experiment, we used a DC boost model MT3608, which has up to 93% efficiency. The model is low cost and small, and the board dimensions are $36 \times 17 \times 14$ mm$^3$ [19].

The measurement unit supplied power to the Arduino Nano microcontroller by connecting a 4-F capacitor in the input voltage (VIN) and ground (GND) pins. At the same time, the wireless protocol nRF24L01 and heart rate sensor were supplied with 3.3 V power, as shown in Figure 1. The Arduino Nano and the wireless protocol are based on the processor ATmega 328P (old bootloader). Further, the Arduino Nano in the receiver was selected because it is small, targeted at low-power consumption, lightweight and low cost. In addition, it is programmed with C++ language. The wireless protocol nRF24L01 is low-power consumption, low cost, small and can send data over long distances of up to 100 meters. The board dimensions are $28.5 \times 15.2$ mm$^2$. The heart rate sensor was linked at low voltage to decrease power consumption. The monitoring unit was supplied by power via Universal Serial Bus (USB) cable from the laptop to power the Arduino Mega and wireless protocol nRF24L01 [20].

The experiment is designed to transfer energy between the Tx and Rx based on UPT. The experimental distance was 4 cm for charging a 4-F super-capacitor. Capacitors have different charging times, as related to Equations (1)–(3) (Figure 3a). The function generator supplied the Tx at a voltage with frequency. The Rx detected a voltage with frequency at 4 cm. The Rx wave was displayed in the storage oscilloscope (model UTD2025CL). Next, the rectifier converted AC to DC, followed by DC converter boosting and super-capacitor action. The digital Avometer describes the relationship between voltage and current in the charging input and output. As a voltage increases, the current decreases to charge. The super-capacitor supplied power for the measurement unit containing the HRM. We selected the Arduino Nano and wireless protocol nRF24L01 at the data rate of 9600 bps to improve the transmission time between the wireless protocol and the microcontroller. Furthermore, the monitoring unit in the HRM involving the Arduino Mega 2560 has the same data rate as the monitoring unit.

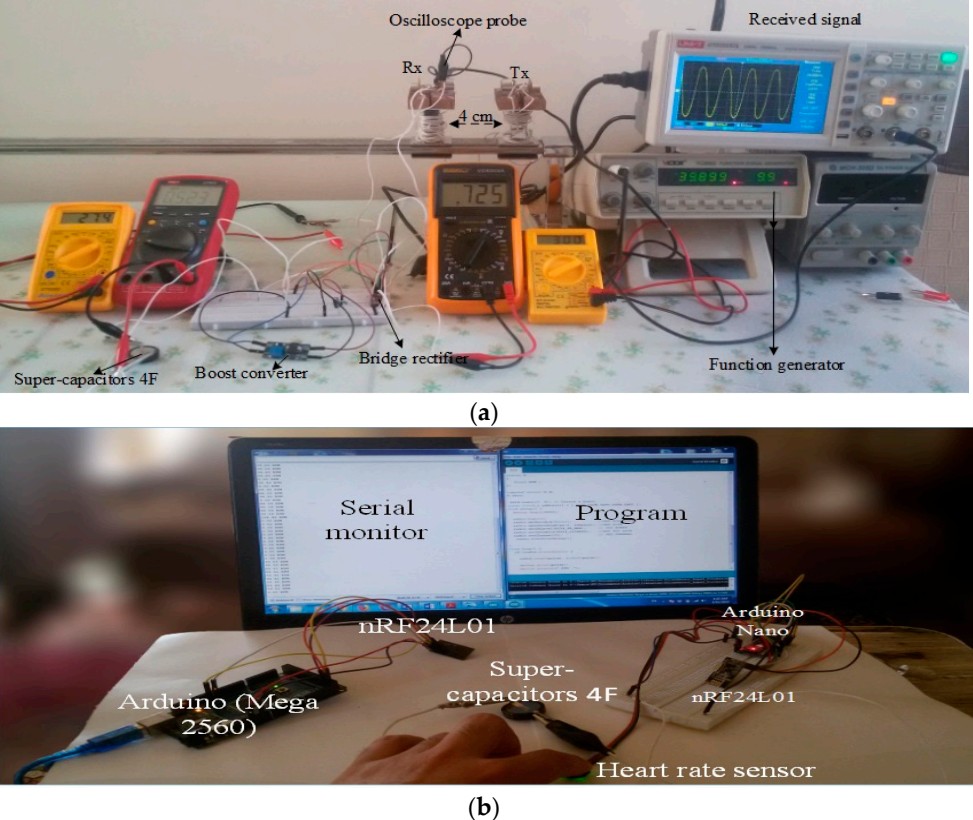

**Figure 3.** A snapshot of the UPT: (**a**) Charging 4-F super-capacitor at 4 cm and (**b**) the coordinator node with its parts for sending data.

In addition, the data rate of wireless protocol nRF24L01 was 250 kbps for transferring data between two models, and it was connected at an input voltage of 3.3 V; the chip-enabled activation of Rx or Tx mode (CE) and SPI chip select (SCN) in the microcontroller on digital pin D7 and D8, respectively. Finally, the laptop was used to demonstrate the result of serial monitoring by using Arduino version 1.8.8. Moreover, heart rate sensor was analog to attach with A0 at the microcontroller (Figure 3b). The heart rate in minutes was calculated using Equation (4) [21].

$$BPM = (60 \times frequency/signal) \tag{4}$$

where *BPM* is beats per minute, *frequency* is the sampling frequency and *signal* is the input signal to the microcontroller to pin A0.

## 5. Evaluation of HRM Based on Statistical Analyses

The device was validated using a BM (i.e., HANSA device). Next, the statistics for the two methods were plotted and compared. The data were obtained from the two systems by wireless protocol nRF24L01. The outcomes indicated reasonable variations in the two systems. The data from the HRM system was validated via statistical analyses. The results are shown in Figure 4. The x-axis shows 100 samples for patients, and the y-axis shows the HRM and BM measurements. A black dash is act BM and a red solid line is act HRM.

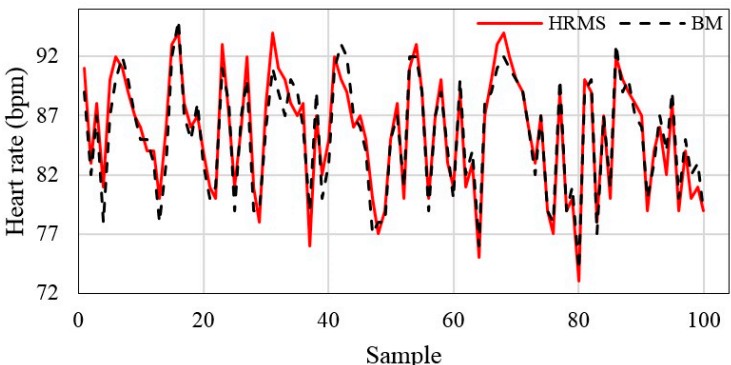

**Figure 4.** Measurement of the heart rate monitoring (HRM) and benchmark (BM) devices.

The data were obtained using the HRM, which was based on UPT and wireless protocol nRF24L01. The proposed HRM was validated by examining and checking the BM. The HRM measurement of the heart rate diverged slightly from that of the BM Figure 5. The two systems were compared and the data obtained by the proposed HRM were examined through statistical analysis (i.e., error test, Bland–Altman test and histogram).

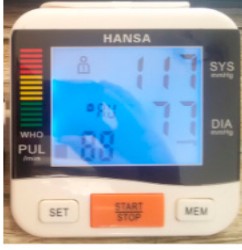

**Figure 5.** The BM device.

### 5.1. Error Test

We examined the error, mean absolute error (MAE), absolute percentage error (APE) and mean absolute percentage error (MAPE) in the heart rate measurements of the HRM with respect to those of

the BM. Figure 6a,b shows the error and APE, respectively, for the heart rate. In Figure 6a, the x-axis shows the BM-measured heart rate in bpm; the y-axis shows the difference between the HRM- and BM-measured heart rates. The figure shows that the absolute error varied over a range of 0–3 with a MAE of 1.26. In Figure 6b, the x-axis shows the BM-measured heart rate in bpm; the y-axis shows the APE. The figure shows that the APE varied over the (0, 3.89) percentage range with a MAPE of 1.479. This result indicates close agreement between the proposed HRM and BM.

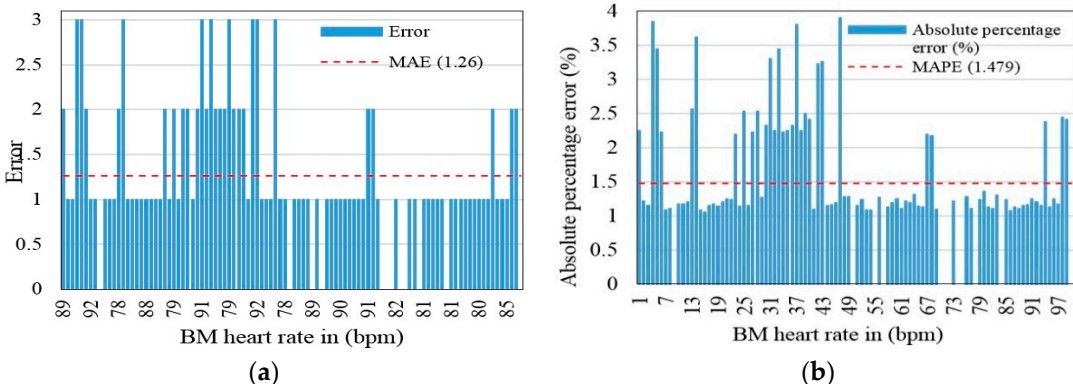

**Figure 6.** Measurement error of heart rate sensor. (**a**) Absolute error with number of samples and (**b**) absolute percentage error (APE) (%) of the proposed HRM relative to the BM.

### 5.2. Bland–Altman Test

The Bland–Altman medical statistical analysis is generally used to compare measurements between the measured and reference values. The differences (i.e., errors) between the heart rate data of the proposed HRM system and the BM are shown as the average ($\mu$) and standard deviation ($\sigma$). The differences were within $\mu \pm 2\sigma$ (97%) limits of agreement (–2.754, 3.15) at the mean difference of the heart rate measurements between the proposed HRM system and the BM. The square points are outside the range; the circle points are inside the range. The average of the system was 0.2 (standard deviation = 1.47) and the width of the 97% limits of agreement was 5.904 (Figure 7).

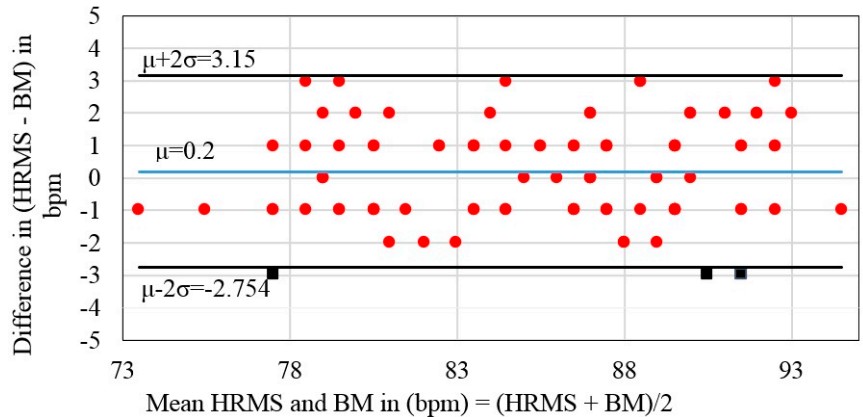

**Figure 7.** Bland–Altman plot for heart rate difference (HRM vs. BM) in beats per minute (bpm).

### 5.3. Histogram

A histogram is a graph that divides continuous data based on frequency distribution and feature representation [22]. The x-axis represents the bases of square shapes that demonstrate the progressive class; the y-axis represents amplitude, i.e., square shapes that illustrate the frequency classes. The bar chart shows a low value, indicating a smaller number of points on the axis, whereas a high value indicates more points on an axis. Therefore, we determined whether the data measured by the

proposed HRM system are compatible with the BM system. The histogram of the heart rate data
(Figure 8) shows peaks of 34 and 27 points in the 89.653 bpm and 89.653 bpm classes for the HRM and
BM device, respectively. These results prove that the data measured by the proposed HRM system are
comparable with that measured by the BM system.

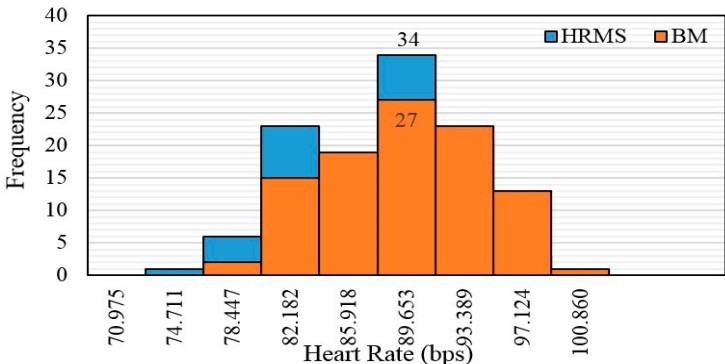

**Figure 8.** Histogram of the heart rate data from the proposed HRM system and BM device.

## 6. Results and Discussion

### 6.1. Charging Super-Capacitors

The different capacitor charging categories are 1 F, 4 F and 8 F for 5 V at different transfer distances
(Figure 9). We used three different super-capacitors with charging at several distances. The x-axis
shows the distance in centimeters; the y-axis shows the time in minutes (Figure 9a). The full charging
time at 4 cm was 200 min, 980 min and 2083 min for 1 F, 4 F and 8 F, respectively. At 10 cm, the charging
time gradually increased to 1813 min, 3078 min and 21,419 min, respectively. The value of resistor to
charging was 10 Ω, and the capacity depended on the capacitors value based on Equations (1)–(3).
When discharging the super-capacitors based on the Arduino Nano and wireless protocol nRF24L01
for the first time, the capacitors were fully charged for 5 V (Figure 9b). The x-axis shows the time in
minutes; the y-axis shows the voltage in volts. The voltage gradually decreased over time to 1.8 V at
12.3 min, 62.2 min and 90.5 min for 1 F, 4 F and 8 F, respectively. The minimum voltage for operating
this system without problems was 1.8 V. A voltage <1.8 V was the cut-off region; a voltage >1.8 V was
the operation region.

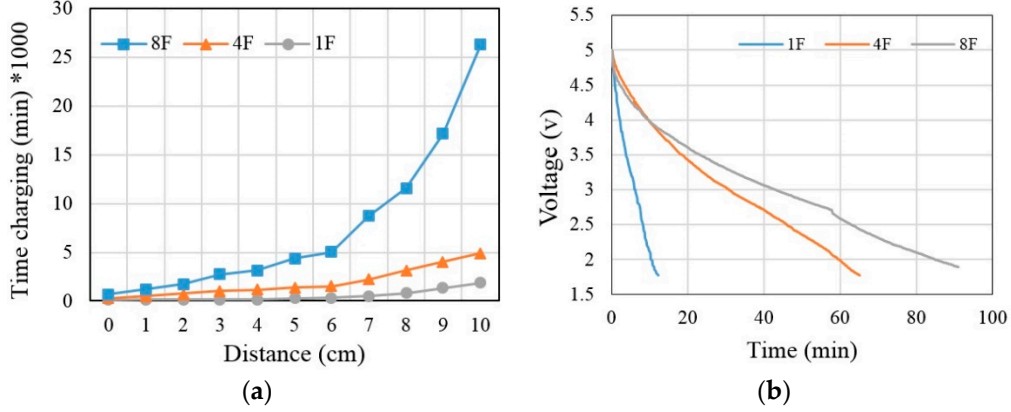

**Figure 9.** Time-based on UPT: (**a**) Charging capacitors and (**b**) discharging capacitors.

### 6.2. Super-Capacitor Voltages and Currents

We performed an experiment for charging super-capacitors at 4 cm based on UPT. The
super-capacitor supplied the Arduino Nano, wireless protocol nRF24L01 and heart rate sensor

for the measurement unit. The capacitor has a voltage and current. When charging the capacitor, the voltage and current are inversely related. In Figure 10a–c, the x-axis shows the time in minutes; the y-axis shows the voltage (dashed black line) and current (solid blue line) in volts and milliampere, respectively. We found that the best voltage to operate the system was 2.8 V, which was the minimum voltage for operating the system without problems. A voltage < 2.8 V was the cut-off region; a voltage >2.8 V was the operation region. However, the time for obtaining this voltage was 133 min, 570 min and 1203 min at 1 F, 4 F and 8 F, respectively. In contrast, the current and time differed based on the capacitor capacity. The current and time were 0.09 mA for 133 min, 0.1514 mA for 570 min and 0.188 mA for 1203 min at 1 F, 4 F and 8 F, respectively.

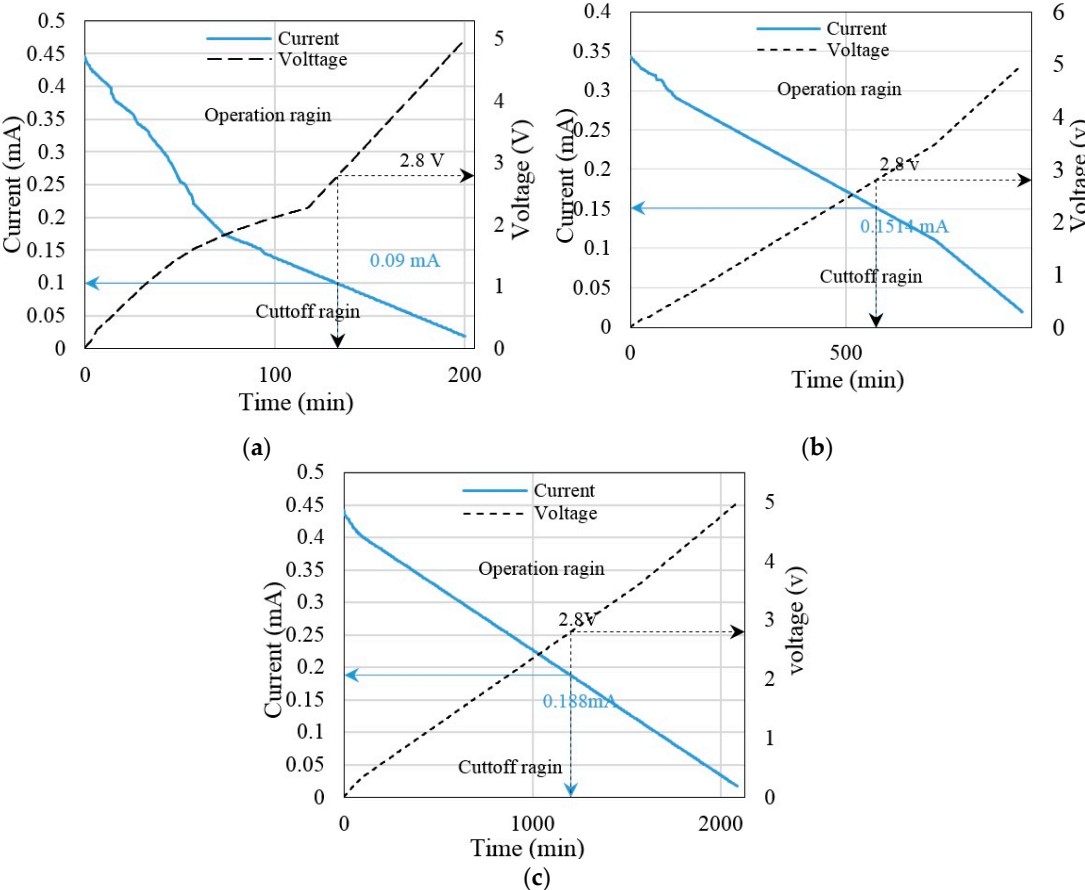

**Figure 10.** Voltage and current at time based on UPT for charging capacitors at 4 cm: (**a**) 1 F, (**b**) 4 F and (**c**) 8 F.

*6.3. Voltage Signal Based on UPT*

When experimental regulation was achieved, the voltage signal with frequency spectrum was measured from the storage oscilloscope (model UTD2025CL) at the Tx. The input voltage at the transmitter was a square wave because it has less loss at a voltage compared with a sinusoidal wave. A drop is about 6 V in the Rx. The input voltage at the transmitter was a square wave for 40 KHz and 10 V$_{\text{peak-to-peak}}$ (Figure 11a). Yellow indicates the input voltage; red indicates the frequency spectrum. Figure 11b shows the signals detected by Rx for the same frequency at 4 cm with the frequency spectrum. The Rx is closed from the Tx, and the signal is approximately a square wave, whereas the signal is a sinusoidal wave at the far sensors. Further, it is pure tone.

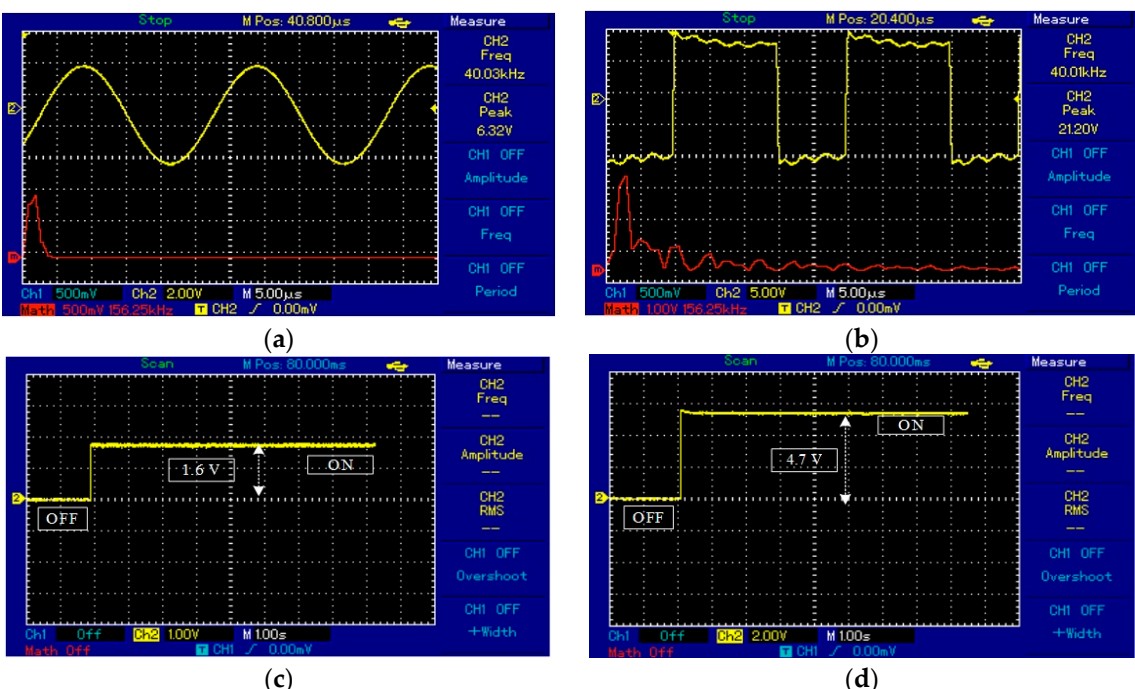

**Figure 11.** The signal of the system at 4 cm: (**a**) Transmitter ultrasonic sensor, (**b**) receiver ultrasonic sensor, (**c**) input signal DC chopper and (**d**) output signal DC chopper.

UPT was based on a distance of 4 cm. In addition, the signal was converted from AC to DC by the rectifier; the output voltage from this distance was 1.6 V at load Figure 11c. In the next stage, the voltage was low, so we used a boost converter to amplify the voltage from 1.6 V to 4.7 V Figure 11d. The x-axis shows the time; the y-axis shows the amplitude. Compared to a direct connection (wire charging), the UPT, based on a distance of 4 cm, has several advantages: (i) It does not require batteries used for low-power devices, (ii) the patient moves freely without being restricted by wires, (iii) it is a light weight-system because it is battery-free, (iv) it does not need a main source of electricity, therefore, it can be used anywhere and (v) the size of the UPT receiver can be minimized to implant inside the human body to supply power to implantable devices.

*6.4. Measurement Transfer Efficiencies and Output Power Based on UPT*

Figure 12 shows the main result of the proposed UPT method. To establish the relationship, the transfer distances in centimeters were plotted on the x-axis, and the transfer efficiencies in percentage were plotted on the y-axis. The transfer efficiency was determined according to equation (5) [3].

$$\text{Transfer efficiency \% } (\eta) \ = \ \frac{\text{output power}}{\text{input power}} \times 100\% \tag{5}$$

The transfer efficiencies are measures of various distances for UPT relay on different loads (Figure 12a). The relationship between the transfer distances (in centimeters) is plotted on the x-axis and the transfer efficiency (%) is plotted on the y-axis. The transfer efficiencies were 86.43%, 76.76% and 58.40% at 1 cm, and were 69.4%, 64.22% and 55.44% at 4 cm. The transfer efficiencies gradually decreased as the distance increased: At 10 cm, transfer efficiencies were 39.39%, 36.80% and 38.91% for loads of 1 KΩ, 1.2 KΩ and 15 KΩ, respectively. The relationship between the transfer distances (in centimeters) is plotted on the x-axis; the output power (in mW) is plotted on the y-axis. The transfer powers are measures for several distances for UPT at different loads (Figure 12b). The output power was 0.206, 0.318 and 0.143 mW at 4 cm. The output power decreased gradually from the peak at 1 cm: At 10 cm, the output power was 0.012, 0.07 and 0.034 mW at 1 KΩ, 1.2 KΩ and 15 KΩ, respectively.

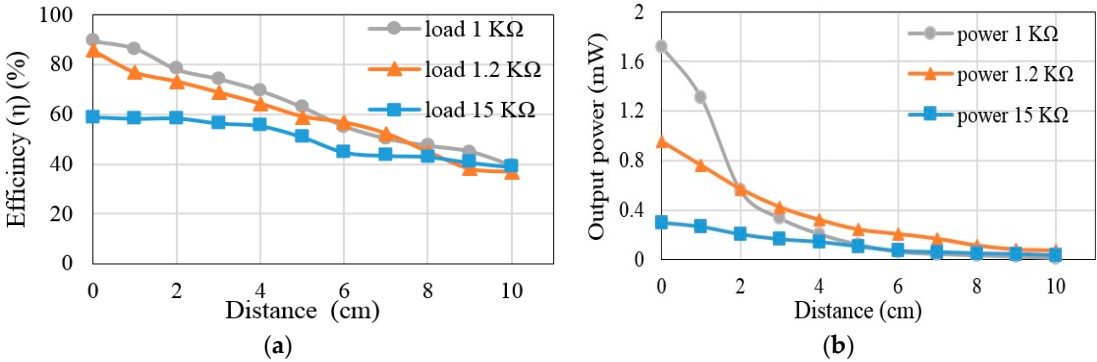

**Figure 12.** The UPT based on load at distance: (**a**) Transfer efficiencies; (**b**) transfer power.

*6.5. Correlation between Transfer Efficiency, Power and Distance*

All parameters in the relationship are shown in a three-dimensional shape. The transfer distances (in centimeters) are plotted on the x-axis; the transfer powers (in mW) are plotted on the y-axis, and the transfer efficiencies (in percentage) are plotted on the z-axis. Dark red, red and orange indicate high values for transfer powers and efficiencies and lows values for transfer distances. Blue indicates low transfer powers and efficiencies and high transfer distances. The loads were 1 KΩ, 1.2 KΩ and 15 KΩ. The transfer powers and efficiencies at 5 cm were 0.119 mW for 62.8%, 0.24 mW for 59.09% and 0.106 mW for 50.93% at 1 KΩ, 1.2 KΩ and 15 KΩ, respectively. Furthermore, the transfer powers and efficiencies decreased gradually as the distance increased at the same loads (Figure 13a–c).

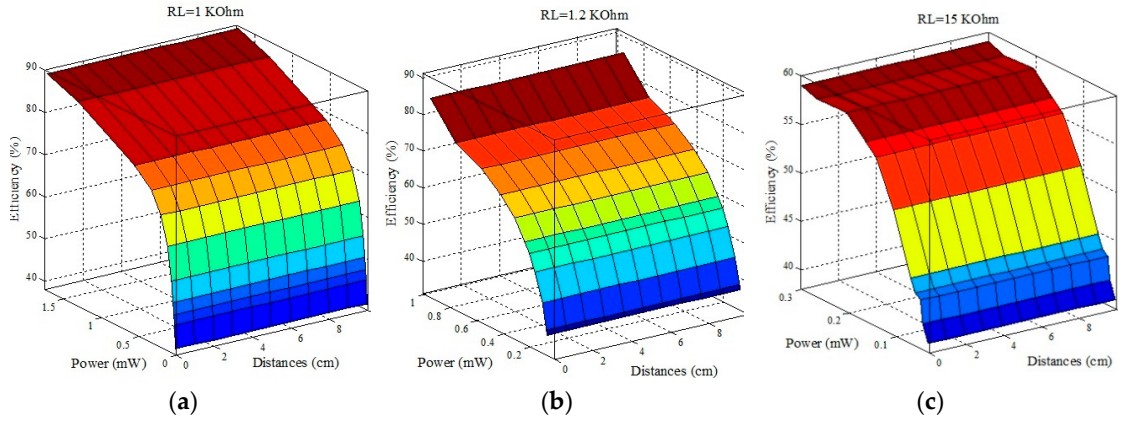

**Figure 13.** The UPT based on load: The three parameters at (**a**) 1 KΩ, (**b**) 1.2 KΩ and (**c**) 15 KΩ.

## 7. Comparison of Results

The previous works were clarified based on UPT and transfer powers, efficiencies and distances at different loads. The bar chart compares the performance metrics of the previous studies in terms of transfer power, efficiency and distance with reference to the present study. Figure 14a shows the transfer power of the references; in the present study, the transfer power was equivalent to 0.318 mW at 4 cm, whereas it was 77 nW in [4]. Figure 14b reviews the transfer efficiencies of the references; here, we observed high transfer efficiency of 69.4% at 4 cm compared with [7,8], which were 5.5% and 1.14%, respectively, at 4 cm. In addition, transfer efficiency was 45% at 0.61 cm in [5]. Figure 14c shows the distance as compared with the previous works, where we compared our results with those of [7,8] based on transfer power and efficiency at 4 cm.

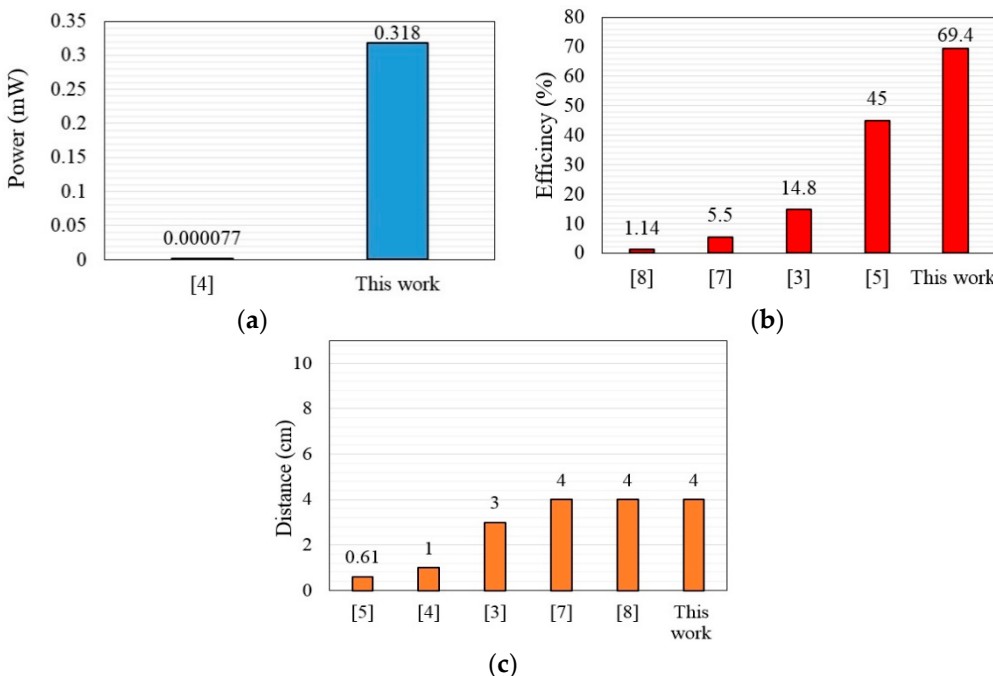

**Figure 14.** Comparison of the present work with related previous studies in terms of: (**a**) Transfer power, (**b**) transfer efficiency and (**c**) distance.

## 8. Conclusions

This paper introduces the design and implementation of UPT for supplying low-power medical devices such as a heart rate sensor. The UPT system was tested on transfer distances of 1–10 cm with 1-, 4-, and 8-F super-capacitors. These super-capacitors were considered to identify the best capacitor that could yield high transfer efficiency and power relative to the other capacitors' values. Therefore, the UPT system achieved 69.4% transfer efficiency and 0.318 mW power when a 4-F super-capacitor was adopted at 4-cm gap between the ultrasound transmitter and receiver. Consequently, the obtained transfer power was adequate for supplying power to the HRM at this distance. However, the transfer power and efficiency decreased as the air gap increased to more than 4-cm. In addition, the experiments show that, at different super-capacitor values, the capacitor charging time increased with distance, especially with a high super-capacitor value such as 8-F. Therefore, we recommend using the proposed UPT with an air gap of <4 cm to obtain adequate voltage in a short time. The results disclosed that the resultant transfer power and efficiency at a 4-cm air gap between the UPT transmitter and receiver was superior to that of previous studies.

Moreover, 97% heart rate measurement accuracy was achieved relative to the benchmark device. The heart rate measurements were validated with statistical analysis. Our results show that this work outperforms previous works in terms of transfer power and efficiency with a 4-cm gap between the ultrasound transmitter and receiver. We can draw that this study demonstrated close agreement between the measured data acquired by the proposed HRM and BM. The results show high measurement accuracy and low MAE. Future work will focus on increasing the transfer power, efficiency and distance of the UPT. In addition, a standalone microcontroller can be used to reduce the power consumption of the measurement system.

**Author Contributions:** Data curation by S.K.G. and M.F.M.; resources and formal analysis by M.F.M.; investigation by S.K.G. and S.L.M.; methodology by M.F.M. and S.K.G.; supervision by S.L.M. and S.K.G.; validation by S.K.G.; visualization by M.F.M.; writing of the original draft by M.F.M. and S.L.M., writing—review and editing by S.K.G. and M.F.M.

**Funding:** This research received no external funding.

**Acknowledgments:** The authors would like to thank Electrical Engineering Technical College, Middle Technical University, Baghdad, Iraq, for their support to conduct the experiments.

**Conflicts of Interest:** The authors declare no conflict of interest.

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
