# Peer review of "Ultrasound Sensor-Based Wireless Power Transfer for Low-Power Medical Devices"

_jlpea, doi:10.3390/jlpea9030020_

Reviewer 1 Report

The manuscript is well written. I commend the authors for their research on alternative power sources for low power medical equipments. Ultrasound is a potential alternative and currently needs an extensive research for it to be able to be used in real time. Manuscript is thorough. The methodology was appropriate. All the methods were elaborate. Tables, figures and all the results were well presented and explained. Conclusions drawn from the results were logical. I would recommend publishing the article after minor format corrections. 

Author Response

Reply to the reviewer as in attached file

Reviewer 2 Report

The paper seems an engineering solution and not an application with a research content.

Authors have:

1.      to highlight the innovativeness and the research content of the paper;

2.      to explain/justify the benefits coming from the adoption of a system characterized by a distance of 4 cm for charging the capacitor with respect to a direct connection;

3.      to finalize results and conclusions to the paper research aim

Moreover, all adopted acronyms have to be introduced and all expressions have to be defined.

Author Response

Reply to the reviewer as in attached file.

Round  2

Reviewer 2 Report

The added part of the conclusion should be re-written because It is composed of no-consequential statements. It seems the collection of stand-alone sentences.

Moreover, expressions 1 and 2 must be reviewed. It is evident that they cannot be currunt and voltage  on a capacitor, given the well-known relationship between the electrical quantities. 

Author Response

The response of the authors as in attached file.
